# Understanding Dropout

**Pierre Baldi**
Department of Computer Science
University of California, Irvine
Irvine, CA 92697
pfbaldi@uci.edu

**Peter Sadowski**
Department of Computer Science
University of California, Irvine
Irvine, CA 92697
pjsadows@ics.uci.edu

## Abstract

Dropout is a relatively new algorithm for training neural networks which relies on stochastically "dropping out" neurons during training in order to avoid the co-adaptation of feature detectors. We introduce a general formalism for studying dropout on either units or connections, with arbitrary probability values, and use it to analyze the averaging and regularizing properties of dropout in both linear and non-linear networks. For deep neural networks, the averaging properties of dropout are characterized by three recursive equations, including the approximation of expectations by normalized weighted geometric means. We provide estimates and bounds for these approximations and corroborate the results with simulations. Among other results, we also show how dropout performs stochastic gradient descent on a regularized error function.

## 1  Introduction

Dropout is an algorithm for training neural networks that was described at NIPS 2012 [7]. In its most simple form, during training, at each example presentation, feature detectors are deleted with probability $q = 1 - p = 0.5$ and the remaining weights are trained by backpropagation. All weights are shared across all example presentations. During prediction, the weights are divided by two. The main motivation behind the algorithm is to prevent the co-adaptation of feature detectors, or overfitting, by forcing neurons to be robust and rely on population behavior, rather than on the activity of other specific units. In [7], dropout is reported to achieve state-of-the-art performance on several benchmark datasets. It is also noted that for a single logistic unit dropout performs a kind of "geometric averaging" over the ensemble of possible subnetworks, and conjectured that something similar may occur also in multilayer networks leading to the view that dropout may be an economical approximation to training and using a very large ensemble of networks.

In spite of the impressive results that have been reported, little is known about dropout from a theoretical standpoint, in particular about its averaging, regularization, and convergence properties. Likewise little is known about the importance of using $q = 0.5$, whether different values of $q$ can be used including different values for different layers or different units, and whether dropout can be applied to the connections rather than the units. Here we address these questions.

## 2  Dropout in Linear Networks

It is instructive to first look at some of the properties of dropout in linear networks, since these can be studied exactly in the most general setting of a multilayer feedforward network described by an underlying acyclic graph. The activity in unit $i$ of layer $h$ can be expressed as:

$$S_i^h(I) = \sum_{l<h} \sum_j w_{ij}^{hl} S_j^l \quad \text{with} \quad S_j^0 = I_j \tag{1}$$

where the variables $w$ denote the weights and $I$ the input vector. Dropout applied to the units can be expressed in the form

$$S_i^h = \sum_{l<h} \sum_j w_{ij}^{hl} \delta_j^l S_j^l \quad \text{with} \quad S_j^0 = I_j \tag{2}$$

where $\delta_j^l$ is a gating 0-1 Bernoulli variable, with $P(\delta_j^l = 1) = p_j^l$ . Throughout this paper we assume that the variables $\delta_j^l$ are independent of each other, independent of the weights, and independent of the activity of the units. Similarly, dropout applied to the connections leads to the random variables

$$S_i^h = \sum_{l<h} \sum_j \delta_{ij}^{hl} w_{ij}^{hl} S_j^l \quad \text{with} \quad S_j^0 = I_j \tag{3}$$

For brevity in the rest of this paper, we focus exclusively on dropout applied to the units, but all the results remain true for the case of dropout applied to the connections with minor adjustments.

For a fixed input vector, the expectation of the activity of all the units, taken over all possible realizations of the gating variables hence all possible subnetworks, is given by:

$$E(S_i^h) = \sum_{l<h} \sum_j w_{ij}^{hl} p_j^l E(S_j^l) \quad \text{for} \quad h > 0 \tag{4}$$

with $E(S_j^0) = I_j$ in the input layer. *In short, the ensemble average can easily be computed by feedforward propagation in the original network, simply replacing the weights $w_{ij}^{hl}$ by $w_{ij}^{hl} p_j^l$.*

## 3 Dropout in Neural Networks

### 3.1 Dropout in Shallow Neural Networks

Consider first a single logistic unit with $n$ inputs $O = \sigma(S) = 1/(1 + ce^{-\lambda S})$ and $S = \sum_1^n w_j I_j$. To achieve the greatest level of generality, we assume that the unit produces different outputs $O_1, \ldots, O_m$, corresponding to different sums $S_1 \ldots, S_m$ with different probabilities $P_1, \ldots, P_m$ ($\sum P_m = 1$). In the most relevant case, these outputs and these sums are associated with the $m = 2^n$ possible subnetworks of the unit. The probabilities $P_1, \ldots, P_m$ could be generated, for instance, by using Bernoulli gating variables, although this is not necessary for this derivation. It is useful to define the following four quantities: the mean $E = \sum P_i O_i$; the mean of the complements $E' = \sum P_i(1 - O_i) = 1 - E$; the weighted geometric mean $(WGM)$ $G = \prod_i O_i^{P_i}$; and the weighted geometric mean of the complements $G' = \prod_i (1 - O_i)^{P_i}$. We also define the normalized weighted geometric mean $NWGM = G/(G + G')$. We can now prove the key averaging theorem for logistic functions:

$$NWGM(O_1, \ldots, O_m) = \frac{1}{1 + ce^{-\lambda E(S)}} = \sigma(E(S)) \tag{5}$$

To prove this result, we write

$$NWGM(O_1, \ldots, O_m) = \frac{1}{1 + \frac{\prod(1-O_i)^{P_i}}{\prod O_i^{P_i}}} = \frac{1}{1 + \frac{\prod(1-\sigma(S_i))^{P_i}}{\prod \sigma(S_i)^{P_i}}} \tag{6}$$

The logistic function satisfies the identity $[1 - \sigma(x)]/\sigma(x) = ce^{-\lambda x}$ and thus

$$NWGM(O_1, \ldots, O_m) = \frac{1}{1 + \prod[ce^{-\lambda S_i}]^{P_i}} = \frac{1}{1 + ce^{-\lambda \sum P_i S_i}} = \sigma(E(S)) \tag{7}$$

*Thus in the case of Bernoulli gating variables, we can compute the $NWGM$ over all possible dropout configurations by simple forward propagation by: $NWGM = \sigma(\sum_1^n w_j p_j I_j)$. A similar result is true also for normalized exponential transfer functions.* Finally, one can also show that the only class of functions $f$ that satisfy $NWGM(f) = f(E)$ are the constant functions and the logistic functions [1].

## 3.2 Dropout in Deep Neural Networks

We can now deal with the most interesting case of deep feedforward networks of sigmoidal units [1], described by a set of equations of the form

$$O_i^h = \sigma(S_i^h) = \sigma(\sum_{l<h}\sum_j w_{ij}^{hl}O_j^l) \quad \text{with} \quad O_j^0 = I_j \tag{8}$$

where $O_i^h$ is the output of unit $i$ in layer $h$. Dropout on the units can be described by

$$O_i^h = \sigma(S_i^h) = \sigma(\sum_{l<h}\sum_j w_{ij}^{hl}\delta_j^l O_j^l) \quad \text{with} \quad O_j^0 = I_j \tag{9}$$

using the Bernoulli selector variables $\delta_j^l$. For each sigmoidal unit

$$NWGM(O_i^h) = \frac{\prod_{\mathcal{N}}(O_i^h)^{P(\mathcal{N})}}{\prod_{\mathcal{N}}(O_i^h)^{P(\mathcal{N})} + \prod_{\mathcal{N}}(1-O_i^h)^{P(\mathcal{N})}} \tag{10}$$

where $\mathcal{N}$ ranges over all possible subnetworks. Assume for now that the $NWGM$ provides a good approximation to the expectation (this point will be analyzed in the next section). Then the averaging properties of dropout are described by the following three recursive equations. First the approximation of means by NWGMs:

$$E(O_i^h) \approx NWGM(O_i^h) \tag{11}$$

Second, using the result of the previous section, the propagation of expectation symbols:

$$NWGM(O_i^h) = \sigma_i^h\left[E(S_i^h)\right] \tag{12}$$

And third, using the linearity of the expectation with respect to sums, and to products of independent random variables:

$$E(S_i^h) = \sum_{l<h}\sum_j w_{ij}^{hl}p_j^l E(O_j^l) \tag{13}$$

Equations 11, 12, and 13 are the fundamental equations explaining the averaging properties of the dropout procedure. The only approximation is of course Equation 11 which is analyzed in the next section. If the network contains linear units, then Equation 11 is not necessary for those units and their average can be computed exactly. In the case of regression with linear units in the top layers, this allows one to shave off one layer of approximations. The same is true in binary classification by requiring the output layer to compute directly the $NWGM$ of the ensemble rather than the expectation. It can be shown that for any error function that is convex up ($\cup$), the error of the mean, weighted geometric mean, and normalized weighted geometric mean of an ensemble is always less than the expected error of the models [1].

Equation 11 is exact if and only if the numbers $O_i^h$ are identical over all possible subnetworks $\mathcal{N}$. Thus it is useful to measure the *consistency* $C(O_i^h, I)$ of neuron $i$ in layer $h$ for input $I$ by using the variance $Var\left[O_i^h(I)\right]$ taken over all subnetworks $\mathcal{N}$ and their distribution when the input $I$ is fixed. The larger the variance is, the less consistent the neuron is, and the worse we can expect the approximation in Equation 11 to be. Note that for a random variable $O$ in [0,1] the variance cannot exceed 1/4 anyway. This is because $Var(O) = E(O^2) - (E(O))^2 \leq E(O) - (E(O))^2 = E(O)(1 - E(O)) \leq 1/4$. This measure can also be averaged over a training set or a test set.

## 4 The Dropout Approximation

Given a set of numbers $O_1, \ldots, O_m$ between 0 and 1, with probabilities $P_1, \ldots, P_M$ (corresponding to the outputs of a sigmoidal neuron for a fixed input and different subnetworks), we are primarily interested in the approximation of $E$ by $NWGM$. The $NWGM$ provides a good approximation because we show below that to a first order of approximation: $E \approx NWGM$ and $E \approx G$. Furthermore, there are formulae in the literature for bounding the error $E - G$ in terms of the consistency (e.g. the Cartwright and Field inequality [6]). However, one can suspect that the $NWGM$ provides even a better approximation to $E$ than the geometric mean. For instance, if the numbers $O_i$ satisfy $0 < O_i \leq 0.5$ (consistently low), then

$$\frac{G}{G'} \leq \frac{E}{E'} \quad \text{and therefore} \quad G \leq \frac{G}{G + G'} \leq E \tag{14}$$

This is proven by applying Jensen's inequality to the function $\ln x - \ln(1 - x)$ for $x \in (0, 0.5]$. It is also known as the Ky Fan inequality [2, 8, 9].

To get even better results, one must consider a second order approximation. For this, we write $O_i = 0.5 + \epsilon_i$ with $0 \leq |\epsilon_i| \leq 0.5$. Thus we have $E(O) = 0.5 + E(\epsilon)$ and $Var(O) = Var(\epsilon)$. Using a Taylor expansion:

$$G = \frac{1}{2}\prod_i \sum_{n=0}^{\infty} \binom{p_i}{n}(2\epsilon_i)^n = \frac{1}{2}\left[1 + \sum_i p_i 2\epsilon_i + \sum_i \frac{p_i(p_i - 1)}{2}(2\epsilon_i)^2 + \sum_{i<j} 4p_i p_j \epsilon_i \epsilon_j + R_3(\epsilon_i)\right] \tag{15}$$

where $R_3(\epsilon_i)$ is the remainder and

$$R_3(\epsilon_i) = \binom{p_i}{3}\frac{(2\epsilon_i)^3}{(1 + u_i)^{3-p_i}} \tag{16}$$

where $|u_i| \leq 2|\epsilon_i|$. Expanding the product gives

$$G = \frac{1}{2} + \sum_i p_i \epsilon_i + (\sum_i \epsilon_i)^2 - \sum_i p_i \epsilon_i^2 + R_3(\epsilon) = \frac{1}{2} + E(\epsilon) - Var(\epsilon) + R_3(\epsilon) = E(O) - Var(O) + R_3(\epsilon) \tag{17}$$

By symmetry, we have

$$G' = \prod_i (1 - O_i)^{p_i} = 1 - E(O) - Var(O) + R_3(\epsilon) \tag{18}$$

where $R_3(\epsilon)$ is the higher order remainder. Neglecting the remainder and writing $E = E(O)$ and $V = Var(O)$ we have

$$\frac{G}{G + G'} \approx \frac{E - V}{1 - 2V} \quad \text{and} \frac{G'}{G + G'} \approx \frac{1 - E - V}{1 - 2V} \tag{19}$$

Thus, to a second order, the differences between the mean and the geometric mean and the normalized geometric means satisfy

$$E - G \approx V \quad \text{and} \quad E - \frac{G}{G + G'} \approx \frac{V(1 - 2E)}{1 - 2V} \tag{20}$$

and

$$1 - E - G' \approx V \quad \text{and} \quad (1 - E) - \frac{G'}{G + G'} \approx \frac{V(1 - 2E)}{1 - 2V} \tag{21}$$

Finally it is easy to check that the factor $(1 - 2E)/(1 - 2V)$ is always less or equal to 1. In addition we always have $V \leq E(1 - E)$, with equality achieved only for 0-1 Bernoulli variables. Thus

$$|E - \frac{G}{G + G'}| \approx \frac{V|1 - 2E|}{1 - 2V} \leq \frac{E(1 - E)|1 - 2E|}{1 - 2V} \leq 2E(1 - E)|1 - 2E| \qquad (22)$$

The first inequality is optimal in the sense that it is attained in the case of a Bernoulli variable with expectation $E$ and, intuitively, the second inequality shows that the approximation error is always small, regardless of whether $E$ is close to 0, 0.5, or 1. *In short, the NWGM provides a very good approximation to E, better than the geometric mean G.* The property is always true to a second order of approximation and it is exact when the activities are consistently low, or when $NWGM \leq E$, since the latter implies $G \leq NWGM \leq E$. Several additional properties of the dropout approximation, including the extension to rectified linear units and other transfer functions, are studied in [1].

## 5 Dropout Dynamics

Dropout performs gradient descent on-line with respect to both the training examples and the ensemble of all possible subnetworks. As such, and with the appropriately decreasing learning rates, it is almost surely convergent like other forms of stochastic gradient descent [11, 4, 5]. To further understand the properties of dropout, it is again instructive to look at the properties of the gradient in the linear case.

### 5.1 Single Linear Unit

In the case of a single linear unit, consider the two error functions $E_{ENS}$ and $E_D$ associated with the ensemble of all possible subnetworks and the network with dropout. For a single input $I$, these are defined by:

$$E_{ENS} = \frac{1}{2}(t - O_{ENS})^2 = \frac{1}{2}(t - \sum_{i=1}^{n} p_i w_i I_i)^2 \qquad (23)$$

$$E_D = \frac{1}{2}(t - O_D)^2 = \frac{1}{2}(t - \sum_{i=1}^{n} \delta_i w_i I_i)^2 \qquad (24)$$

We use a single training input $I$ for notational simplicity, otherwise the errors of each training example can be combined additively. The learning gradient is given by

$$\frac{\partial E_{ENS}}{\partial w_i} = -(t - O_{ENS})\frac{\partial O_{ENS}}{\partial w_i} = -(t - O_{ENS})p_i I_i \qquad (25)$$

$$\frac{\partial E_D}{\partial w_i} = -(t - O_D)\frac{\partial O_D}{\partial w_i} = -(t - O_D)\delta_i I_i = -t\lambda_i I_i + w_i \delta_i^2 I_i^2 + \sum_{j \neq i} w_j \delta_i \delta_j I_i I_j \qquad (26)$$

The dropout gradient is a random variable and we can take its expectation. A short calculation yields

$$E\left(\frac{\partial E_D}{\partial w_i}\right) = \frac{\partial E_{ENS}}{\partial w_i} + w_i p_i (1 - p_i) I_i^2 \frac{\partial E_{ENS}}{\partial w_i} + w_i I_i^2 Var(\delta_i) \qquad (27)$$

*Thus, remarkably, in this case the expectation of the gradient with dropout is the gradient of the regularized ensemble error*

$$E = E_{ENS} + \frac{1}{2}\sum_{i=1}^{n} w_i^2 I_i^2 Var(\delta_i) \qquad (28)$$

The regularization term is the usual weight decay or Gaussian prior term based on the square of the weights to prevent overfitting. Dropout provides immediately the magnitude of the regularization term which is adaptively scaled by the inputs and by the variance of the dropout variables. Note that $p_i = 0.5$ is the value that provides the highest level of regularization.

## 5.2 Single Sigmoidal Unit

The previous result generalizes to a sigmoidal unit $O = \sigma(S) = 1/(1 + ce^{-\lambda S})$ trained to minimize the relative entropy error $E = -(t \log O + (1 - t) \log(1 - O))$. In this case,

$$\frac{\partial E_D}{\partial w_i} = -\lambda(t - O)\frac{\partial S}{\partial w_i} = -\lambda(t - O)\delta_i I_i \qquad (29)$$

The terms $O$ and $I_i$ are not independent but using a Taylor expansion with the $NWGM$ approximation gives

$$E\left(\frac{\partial E_D}{\partial w_i}\right) \approx \frac{\partial E_{ENS}}{\partial w_i} + \lambda\sigma'(S_{ENS})w_i I_i^2 Var(\delta_i) \qquad (30)$$

with $S_{ENS} = \sum_j w_j p_j I_j$. *Thus, as in the linear case, the expectation of the dropout gradient is approximately the gradient of the ensemble network regularized by weight decay terms with the proper adaptive coefficients.* A similar analysis, can be carried also for a set of normalized exponential units and for deeper networks [1].

## 5.3 Learning Phases and Sparse Coding

During dropout learning, we can expect three learning phases: (1) At the beginning of learning, when the weights are typically small and random, the total input to each unit is close to 0 for all the units and the consistency is high: the output of the units remains roughly constant across subnetworks (and equal to 0.5 with $c = 1$). (2) As learning progresses, activities tend to move towards 0 or 1 and the consistency decreases, i.e. for a given input the variance of the units across subnetworks increases. (3) As the stochastic gradient learning procedure converges, the consistency of the units converges to a stable value.

Finally, for simplicity, assume that dropout is applied only in layer $h$ where the units have an output of the form $O_i^h = \sigma(S_i^h)$ and $S_i^h = \sum_{l<h} w_{ij}^{hl} \delta_j^l O_j^l$. For a fixed input, $O_j^l$ is a constant since dropout is not applied to layer $l$. Thus

$$Var(S_i^h) = \sum_{l<h}(w_{ij}^{hl})^2(O_j^l)^2 p_j^l(1 - p_j^l) \qquad (31)$$

under the usual assumption that the selector variables $\delta_j^l$ are independent of each other. Thus $Var(S_i^h)$ depends on three factors. Everything else being equal, it is reduced by: (1) Small weights which goes together with the regularizing effect of dropout; (2) Small activities, which shows that dropout is not symmetric with respect to small or large activities. Overall, dropout tends to favor small activities and thus sparse coding; and (3) Small (close to 0) or large (close to 1) values of the dropout probabilities $p_j^l$. Thus values $p_j^l = 0.5$ maximize the regularization effect but may also lead to slower convergence to the consistent state. Additional results and simulations are given in [1].

## 6 Simulation Results

We use Monte Carlo simulation to partially investigate the approximation framework embodied by the three fundamental dropout equations 11, 12, and 13, the accuracy of the second-order approximation and bounds in Equations 20 and 22, and the dynamics of dropout learning. We experiment with an MNIST classifier of four hidden layers (784-1200-1200-1200-1200-10) that replicates the results in [7] using the Pylearn2 and Theano software libraries[12, 3]. The network is trained with a dropout probability of $0.8$ in the input, and $0.5$ in the four hidden layers. For fixed weights and a fixed input, 10,000 Monte Carlo simulations are used to estimate the distribution of activity $O$ in each neuron. Let $O^*$ be the activation under the deterministic setting with the weights scaled appropriately.

The left column of Figure 1 confirms empirically that the second-order approximation in Equation 20 and the bound in Equation 22 are accurate. The right column of Figure 1 shows the difference between the true ensemble average $E(O)$ and the prediction-time neuron activity $O^*$. This difference grows very slowly in the higher layers, and only for active neurons.

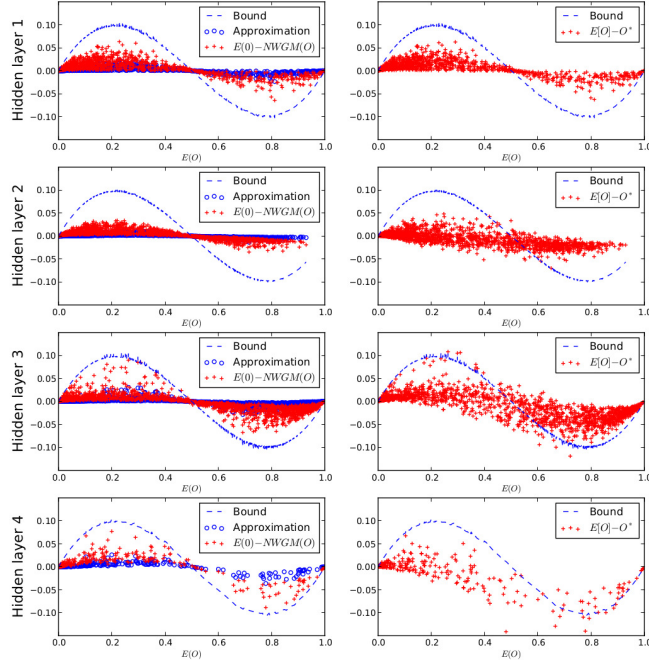

Figure 1: Left: The difference $E(O) - NWGM(O)$, it's second-order approximation in Equation 20, and the bound from Equation 22, plotted for four hidden layers and a typical fixed input. Right: The difference between the true ensemble average $E(O)$ and the final neuron prediction $O^*$.

Next, we examine the neuron consistency during dropout training. Figure 2a shows the three phases of learning for a typical neuron. In Figure 2b, we observe that the consistency does not decline in higher layers of the network.

One clue into how this happens is the distribution of neuron activity. As noted in [10] and section 5 above, dropout training results in sparse activity in the hidden layers (Figure 3). This increases the consistency of neurons in the next layer.

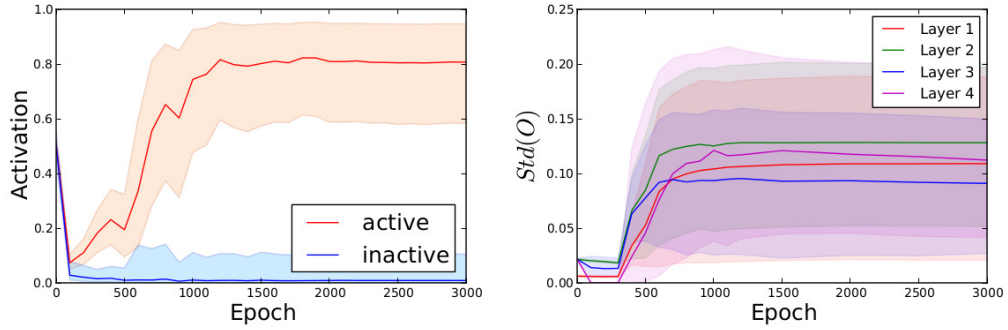

(a) The three phases of learning. For a particular input, a typical active neuron (red) starts out with low variance, experiences a large increase in variance during learning, and eventually settles to some steady constant value. In contrast, a typical inactive neuron (blue) quickly learns to stay silent. Shown are the mean with 5% and 95% percentiles.

(b) Consistency does not noticeably decline in the upper layers. Shown here are the mean $Std(O)$ for active neurons ($0.1 < O$ after training) in each layer, along with the 5% and 95% percentiles.

Figure 2

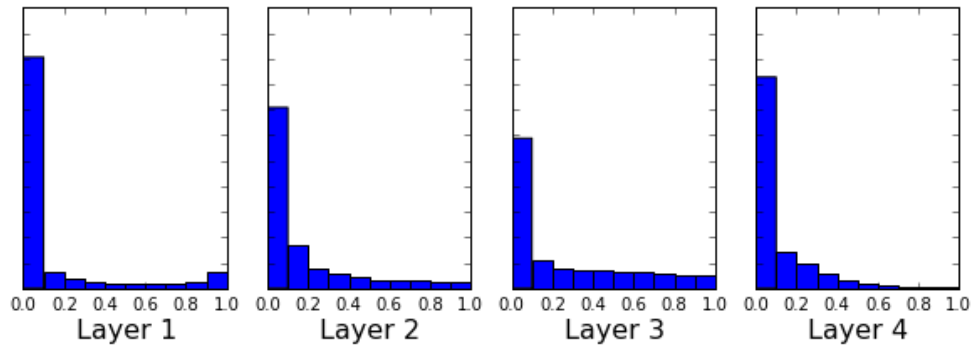

Figure 3: In every hidden layer of a dropout trained network, the distribution of neuron activations $O^*$ is sparse and not symmetric. These histograms were totalled over a set of 100 random inputs.

## Footnotes

[1]Given the results of the previous sections, the network can also include linear units or normalized exponential units.

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
