[Reviews · NeurIPS 2013]

Submitted by Assigned_Reviewer_4

This work aims to shed light to the novel neural network algorithm of dropout, but grounding it to theory and through this provide guidance of how relevant parameters should be chosen to achieve a good performance.

Quality: The quality of the work is high, the quality of the manuscript can be however, improved: there are several typos, or missing symbol definitions. A summary of the findings at the end would also enhance the readability of the manuscript.

Clarity: The paper is overall clearly written subject to typos, definitions and summary missing (see above). A non-exhaustive list of suggested corrections is given below:

-Introduction, line 7, there may be a typo/ ambiguous
-Eq (1): l needs to be defined
-3.1, n needs to be defined. Is stochasticity is S due to dropout? Please explain m=2^n (sems to be the number of different inputs in the network). Typo in equation 7
-5.1 typo in eq 25, typo right after eq 28 (an extra fullstop)
-5.2 first line: in the definition of the sigmoidal a + is missing.
-the caption layout of the figures is non consistent
-references in the text start from number [6]
-A summary /conclusions section is missing


Originality: To the best of my knowledge, the paper is original, providing answers to some key questions regarding the setup of a network with dropout.

Significance: In my view the article is significant, and I was wondering if parallels could be drawn between the dropout algorithm, applied to the connecting weights rather than the nodes, with the release probability of biological synapses, as those may be unreliable, effectively implementing a “dropout” mechanism.
Summary: This is an interesting work that helps grounding the dropout algorithm to solid mathematical foundations. The manuscript though has a few typos, and its readability can be improved.

Submitted by Assigned_Reviewer_5

Summary.

Dropout is an algorithm that was recently introduced by Krizhevsky et al. This paper introduces a general formalism for studying and understanding dropout. It provides a useful analysis of what exactly dropout does, and how it regularizes a deep network.

Quality.

Very good writeup. The paper first summarizes what dropout is, and how its formulation is exact for linear networks. It then goes on by demonstrating what happens (and what approximations occur) with neural networks. It concludes with simulation results which empirically confirm most of the theory introduced in the previous sections.

The authors also describe the 3 typical learning phases that occur while using dropout, and also support these claims with simulation data. This is a quite interesting analsyis.

Disclaimer: I haven't gone through all the math,

Clarity.

Very clear overall.

Originality / Significance.

The Dropout algorithm itself was quite original; I expect to see quite a few papers (at NIPS) trying to understand and analyze it. Nevertheless, given the impact that dropout had, it is very important to understand why and how it works, and this paper can draw attention, given its usefulness.
Summary: A good paper whose aim is to understand what and how dropout works. This type of work is needed, and nicely excecuted here.

Submitted by Assigned_Reviewer_6

This paper presents a mathematical analysis of the dropout procedure in deep neural networks. As far as I know, this is the first attempt to prove the some what heuristically used dropout procedure. There have been some suggestions in prior work (at least for the shallow case) that dropout performs some form of an averaging (geometric to be precise). But this is the first attempt to prove this property for deep neural networks and show that the normalized version of the weighted geometric version provides better approximations than the traditional geometric average. In particular, the three equations 11, 12, 13 are important contributions of this paper and will have a greater impact on future work on deep neural networks, especially in their theoretical analysis.

Minor comments
-----------------------
Eq. 7: factor c should appear before exp term in one before the last term. Missing closing bracket at the end of the expression.

Extract parenthesis in Eq 25 after the differential

Perhaps lambda_i in Eq. 26 is actually delta_i?

It wasn't clear to me the claim about p_i = 0.5 giving the highest level of regularization. Authors could clarify this point in bit more detail in the paper because it is an important observation that justifies the current heuristic in the dropout.

Figure 3 (page 8) appears too small. You can organize the figures in page 8 in a better way and avoid the space between the two figures.
Summary: This paper presents a mathematical analysis of the dropout procedure in deep neural networks. As far as I know, this is the first attempt to prove the some what heuristically used dropout procedure. There have been some suggestions in prior work (at least for the shallow case) that dropout performs some form of an averaging (geometric to be precise). But this is the first attempt to prove this property for deep neural networks and show that the normalized version of the weighted geometric version provides better approximations than the traditional geometric average. In particular, the three equations 11, 12, 13 are important contributions of this paper and will have a greater impact on future work on deep neural networks, especially in their theoretical analysis.
Author Feedback

Author rebuttal: 